# Connexin and Pannexin Large-Pore Channels in Microcirculation and Neurovascular Coupling Function

**DOI:** 10.3390/ijms23137303

**Published:** 2022-06-30

**Authors:** Pía C. Burboa, Mariela Puebla, Pablo S. Gaete, Walter N. Durán, Mauricio A. Lillo

**Affiliations:** 1Department of Pharmacology, Physiology and Neuroscience, New Jersey Medical School, Rutgers University, 185 South Orange Avenue, Newark, NJ 07103, USA; pb588@njms.rutgers.edu (P.C.B.); duran@njms.rutgers.edu (W.N.D.); 2Departamento de Morfología y Función, Facultad de Salud y Ciencias Sociales, Sede Santiago Centro, Universidad de las Américas, Avenue República 71, Santiago 8370040, Chile; mpueblac@gmail.com; 3Department of Physiology and Membrane Biology, University of California at Davis, Davis, CA 95616, USA; psgaete@ucdavis.edu; 4Rutgers School of Graduate Studies, 185 South Orange Avenue, Newark, NJ 07103, USA

**Keywords:** microcirculation, gap junction, hemichannels

## Abstract

Microcirculation homeostasis depends on several channels permeable to ions and/or small molecules that facilitate the regulation of the vasomotor tone, hyperpermeability, the blood–brain barrier, and the neurovascular coupling function. Connexin (Cxs) and Pannexin (Panxs) large-pore channel proteins are implicated in several aspects of vascular physiology. The permeation of ions (i.e., Ca^2+^) and key metabolites (ATP, prostaglandins, D-serine, etc.) through Cxs (i.e., gap junction channels or hemichannels) and Panxs proteins plays a vital role in intercellular communication and maintaining vascular homeostasis. Therefore, dysregulation or genetic pathologies associated with these channels promote deleterious tissue consequences. This review provides an overview of current knowledge concerning the physiological role of these large-pore molecule channels in microcirculation (arterioles, capillaries, venules) and in the neurovascular coupling function.

## 1. Introduction

The cardiovascular system is constituted by the heart, arteries, arterioles, capillaries, postcapillary venules, venules, and veins. The heart pumps the blood to all tissues, allowing the continuous supply of oxygen and crucial metabolites to support tissue homeostasis. Elastic arteries form the systemic circulation, such as the aorta, work under high pressure, and carry the blood to the arterioles. Arterioles play a crucial role in regulating blood flow and peripheral resistance. The blood then flows into the capillary bed to supply nutrients and meet the metabolic demands of tissues. Subsequently, the blood returns to the heart via venules and veins.

In the microcirculation, which includes arterioles, capillaries, and postcapillary venules, connexin and pannexin channels play a vital role in maintaining the activity of microcirculation homeostasis. This review will focus on Cxs and Panxs large channels in microcirculation and neurovascular coupling regulation.

### 1.1. Connexin Channels

Connexin channels form two different functional channels: gap junctional (GJ) channels and hemichannels. GJ channels directly connect the cytoplasm of adjacent cells [1,2,3]. Connexin (Cxs) hemichannels (also termed connexons) are the hexameric single-membrane subunits of GJ contributed by each cell (Figure 1). GJ channels and Cxs hemichannels are permeable to ions and small molecules (<1.5 kDa) [4,5,6,7,8,9,10]. Twenty connexin isoforms are expressed in mammalian cells, and each connexin isoform is referred to according to its molecular weight (i.e., Cx43 has a molecular weight of 43 kDa) [11]. Tissues and cells, including the vascular microcirculation, often express more than one connexin isoform [12,13,14]. Hemichannels can be organized by an association of one (homomeric channels) or a combination (heteromeric channels) of additional connexin isoforms, which normally differ in unitary conductance, ion/molecule permeability, and regulation [4,6,8,11,15,16]. The activity of connexin proteins is regulated by several factors including: voltage, pH, carbon monoxide, and extracellular–intracellular Ca^2+^ [4,11,17,18,19,20,21,22]. Connexin post-translational modifications such as phosphorylation or S-nitrosylation play a role in keeping the function and homeostasis of the tissue [23,24,25,26,27]. Cxs hemichannels also promote ion permeation (i.e., Ca^2+^) [6,9,15,28,29,30] and molecule release (ATP, D-serine, prostaglandins, nitric oxide, etc.) to the milieu, playing a vital role in cell function activity [15,31,32,33,34,35].

Under some circumstances, mutant, and even wild type (WT) hemichannels contribute to pathologies. For example, several connexin mutations that cause human pathologies result from an exacerbated hemichannel opening at the plasma membrane, with deleterious consequences [39,40,41,42,43,44,45]. Similarly, in ischemia and inflammation, there is strong evidence that the opening of hemichannels enhances tissue damage in astrocytes, hepatocytes, and cardiac and skeletal myocytes [5,15,46,47,48,49,50,51,52,53]. In contrast to pathology, the physiological role of connexin hemichannels is not well defined.

### 1.2. Pannexin Channels

Panxs, similar to Cxs hemichannels, represent another family of large pore channels. The major structural difference is that pannexin (i.e., Panx-1) channels are heptameric channels [54,55,56] compared to connexins, that are hexameric channels [36,37,38,57,58,59] (Figure 1). Three members of this protein family have been described (Panx-1, Panx-2, and Panx-3) [60]. Evidence reveals that Panx-1 channels cannot anchor with additional pannexins from adjacent cells probably because of several glycosylations located in extracellular loops [61]. However, recent studies demonstrated that Panx-1 overexpressed in HeLa cells form GJ channels [62]. Nevertheless, it is still unclear whether or not Panx-1 proteins dock with adjacent cells under physiological conditions.

Panx-1 is the most studied pannexin because its expression is detected in several cells and tissues. Similar to connexin hemichannels, Panx-1 facilitates the release of small metabolites to extracellular space, for instance, purines (i.e., ATP), that afterward signal through the activation of membrane receptors (i.e., purinergic receptors) in neighboring cells or even at some distance [63,64,65].

## 2. Connexin Expression in Arterioles

Connexin and pannexin channels are usually evaluated through the manipulation of rodent models or human primary cells. Here, we briefly summarize the expression of connexin proteins in peripheral small vessels (Table 1).

Interactions of one or more connexin isoforms (Cx37, Cx40, Cx43, and Cx45) expressed in the vessel wall [12,66,72,73,74,75,76] mediate intercellular communication via gap junctions in the vascular system. The pattern of connexin expression in the vasculature is variable and depends on vessel size, vascular territory, and species [66,67,73,76,77] (Table 1). Cx37, Cx40, and Cx43 are expressed in endothelial cells (ECs) and smooth muscle cells (SMCs) [12,66,68,75,78,79,80]. Cx45 has been observed in the SMCs of cerebral vessels [81,82,83,84]. The goal of the cited studies [12,66,68,73,75,76,77,78,79,80,81,82,83,84] was to evaluate the role of gap junction channels in microcirculation homeostasis. However, the direct contribution of Cxs hemichannels to the regulation of the vasomotor tone was not evaluated and is incompletely understood. 

ECs in resistance arteries are a critical component in the tonic regulation of vascular homeostasis by mainly the Ca^2+^-dependent production of vasorelaxation signaling as nitric oxide (NO) and prostaglandins [69,85,86,87]. However, another NO- and prostaglandin-independent response has been observed only in arterioles, which is associated with SMCs hyperpolarization [12,66,68,75,78,79,80]. This smooth muscle hyperpolarization signaling produced in ECs is transmitted to SMCs, leading to relaxation by the consequent reduction in the open probability of L-type voltage-dependent Ca^2+^ channels. The contraction intensity of SMCs is determined by the intracellular Ca^2+^ concentration and the Ca^2+^ sensitivity of the contractile apparatus. L-type voltage-dependent Ca^2+^ channels of vascular smooth muscle play a central role in the control of the vasomotor tone by changes in the membrane potential: depolarization produces a Ca^2+^ influx that leads to vasoconstriction, while hyperpolarization results in a decrease in the intracellular Ca^2+^ concentration that leads to vasodilation. In this context, connexin gap junction channels have an essential role in the regulation of the blood flow distribution because these channels allow a direct cell-to-cell communication [12,66,68,75,78,79,80] and participate in the regulation of the vessels diameter. Therefore, cell communication in the vessel wall has emerged as a critical and relevant signaling pathway to coordinate vascular wall function in resistance arteries by the radial (among ECs and SMCs) and longitudinal (along the vessel length) conduction of vasomotor signals.

ECs and SMCs are physically divided by the internal elastic lamina in arterioles; these cells are in contact via cell promontories that skewer the internal elastic lamina and match the other cell type at points comprehended as myoendothelial junctions [12,66,68,73,75,76,77,78,79,80,81,82,83,84]. These points of connection seem to form approvingly specialized subcellular signaling microdomains, and gap junctions located at myoendothelial junctions (i.e., myoendothelial gap junctions) supply an essential pathway for a key regulation of the vasomotor tone through the radial transmission of current and molecules as we described above (i.e., Ca^2+^ and small signaling metabolites). However, whether connexin hemichannels facilitate the Ca^2+^ influx or metabolite release through the endothelial or smooth muscle membrane is not completely understood.

### Role of Connexin Protein Function in Arterioles

As we described above, there is more than one connexin type in the vascular wall from small vessels (Table 1). Here we describe the participation and the main contribution of each connexin to the vascular homeostasis function.

Cx40 knockout mice show irregular arteriolar vasomotion affecting gap junction communication in ECs and a reduction of conducted vasodilation along the vessel [12,88]. The cellular pathway of conducted vasomotor signals depends on the cell type that starts the vasomotion response. For example, vasoconstrictor agonist responses (i.e., phenylephrine [PE]), which activates SMCs, are invariably conducted by SMCs, not by the endothelium layer [89,90]. Vasodilator signals evoked by acetylcholine (ACh) or bradykinin (BK) are spread exclusively by ECs in feed arteries [91,92] or by both SMCs and ECs in arterioles [89,90], which shows that the cellular conduction pathway depends on the functional location of the particular vessel in the microvascular network. However, the cellular path of vasodilator signals may also depend on the stimulus that initiated the response because, in contrast to ACh, selective damage of the endothelium precludes the vasodilation induced by BK in arterioles [90,93].

Deletion of Cx37 does not affect vasomotor function or arterial blood pressure [12], suggesting that Cx37 is not involved in the vasomotor tone. The endothelial-specific knockout of Cx43 (EC-Cx43^−/−^), promotes hypotension by reducing gap junction communication between ECs and SMCs [94]. The molecular mechanisms of this hypotensive phenotype are elusive. It is most likely that the disruption of Cx43 GJ channels increases eNOS activity, promoting hypotension. The colocalization of eNOS and myoendothelial Cx43 GJ channels coordinate the vasomotor tone [95]. Therefore, a reduction of Cx43 GJ channels’ formation in the myoendothelial space promotes more cytosolic eNOS activation, exacerbating endothelial NO production, leading to a hypotensive phenotype. In ECs, eNOS is predominantly localized in caveolae and released by inhibition via caveolin-1 (Cav-1) upon the stretching of the vascular wall (as reviewed in [96]). In Cav-1 knockout (KO) mice, eNOS is constitutively activated, leading to enhanced NO production. Saliez et al. showed that these mice have fewer gap junctions in vessels due to a lower expression of Cxs 37, 40, and 43, and the fact that TRPV4 channels and Cxs colocalize with Cav-1 in caveolae [97]. Cav-1 KO mice were reported to develop cardiac hypertrophy, pulmonary hypertension, and systemic hypotension [98]. Since this knockout eliminates Cx43 in both GJ channels and hemichannels, the specific contribution of Cx43 hemichannels in regulating the vasomotor tone is not clear. Cxs hemichannels function has not been explored directly in arterioles, in contrast with other tissues or cells where Cx43 hemichannels in physiological conditions regulate metabolite release (ATP, D-serine), electrical activity behavior (i.e., action potentials, resting membrane potential), or the potential redox balance in tanycytes, cardiac cells, astrocytes, neurons, and mouse lenses, respectively [15,29,63,64,65,99,100,101].

The endothelium is critical in mediating vascular function by the Ca^2+^-dependent generation of vasodilator signals such as nitric oxide (NO) and prostaglandins [69,85,86,87]. Although NO is the major endothelium-dependent vasodilator signal in large conduit vessels, NO or prostaglandin production inhibition solely attenuates the vasorelaxation initiated by endothelium-dependent vasodilators in small resistance arteries [86]. The vasodilatory process in arterioles involves the hyperpolarization of SMCs [13,102,103,104]. This process is closely associated with the endothelium-derived hyperpolarizing factor (EDHF), which plays a fundamental role controlling the vasomotor tone in small arteries [13,102,103,104].

Several EDHF candidates have been proposed, such as K^+^ ions [105], NO, epoxyeicosatrienoic acids (EETs) [106,107], hydrogen peroxide [108], and C-type natriuretic peptide (CNP) [109,110], but the identity of EDHF has not been established. EDHF-mediated vasodilation is paralleled by the hyperpolarization of the endothelium. Endothelial hyperpolarization signaling is completely prevented by the simultaneous inhibition of Ca^2+^-activated K^+^ channels (K_Ca_) of small (SK_Ca_) and intermediate conductance (IK_Ca_) [103,111,112], indicating that K_Ca_ channels mediate this endothelial electrical signal. These K^+^ channels are expressed in ECs but not in SMCs [112,113]. The endothelial hyperpolarization triggered directly by K^+^ channel activation is called endothelium-derived hyperpolarization (EDH) [114]. As the longitudinal and radial transmission of endothelial hyperpolarization depends on connexin gap junction communication between ECs and SMCs to evoke vessel dilation, EDH-associated vasodilatory signaling is prevented by pharmacological approaches that block both GJ channels and hemichannels [115,116,117]. As this methodology blocks GJ channels and hemichannels, these studies [115,116,117] do not distinguish the effects of blocking the connexin GJ channels from the effects of blocking the Cxs-hemichannels with regards to endothelium hyperpolarization.

## 3. Pannexin Proteins in Arterioles

Panx-1 is almost ubiquitously expressed in murine ECs and SMCs of large and small vessels, while Panx-3 is only found in arterioles [65,70,71,118,119]. In addition, Panx-2 is expressed in SMCs of the pulmonary artery of mice and in the SMCs of the rat middle cerebral artery (MCA) [71].

Several studies about pannexin function, focusing mainly on Panx-1, have been performed in SMCs. For example, Panx-1 and the α1-adrenergic receptor (AR) are coupled in the regulation of vasoconstriction processes [119,120,121]. SMCs-specific Panx-1 deletion, as well as the use of multiple pannexin inhibitors (such as mimetic peptides, trovafloxacin, and spironolactone), blunts the noradrenaline- and phenylephrine-mediated vasoconstriction of resistance arteries [122,123].

These studies strongly support the concept that smooth muscle Panx-1 channels regulate the vasomotor tone in arterioles. In addition, the regulation of Panx-1 could play a key role in the vasomotor tone in endothelial cells. NO potently inhibits endothelial Panx-1 channels by S-nitrosylation at amino acids cysteine 40 and cysteine 346 to prevent channel opening and ATP release [124], which could be associated with peripheral resistance homeostasis.

Independent of the NO signaling pathway, the activation of endothelial Panx-1 channel-initiated purinergic signaling mediates EDH-dependent vasodilation by the endothelium-dependent vasodilator ACh [125]. Nevertheless, although the EDH-mediated vasodilator pathway is reduced in Panx-1 knockout mice, the NO-dependent vasodilator component is enhanced in these animals, indicating that endothelial Panx-1 may be involved in the tonic regulation of NO production by eNOS. Similar results have been found in mesenteric rat vessels, where general blockers of Panx-1 channels promote endothelial superoxide production by activating endothelial NADP(H)oxidase [70]. Endothelial TTX-sensitive Na_v_ channels and Ca_v_3.2 channels mediate endothelial depolarization and this superoxide production, promoting an increase in NO-mediated vasorelaxation via NADPH oxidase-dependent eNOS phosphorylation at serine 1177 [70]. On the other hand, the prolonged activation of Panx-1 channels by Calcitonin Gene-Related Peptide (CGRP) receptor stimulation leads to a Panx-1-formed channel opening and the ensuing superoxide-dependent connexin-based hemichannel activation exclusively in ECs [69]. The lengthy opening of endothelial Panx-1 channels by perivascular sensory nerves results in the progressive inhibition of NO production by reducing the eNOS phosphorylation status at serine 1177 [69]. These studies, [69] and [70], indicate that endothelial Panx-1 signaling seems to be more complex and is associated to regulate NO production through the eNOS phosphorylation status by a NADPH oxidase/O_2_^⋅–^-mediated signaling. These results also indicate that Panx-1 may participate in the endothelium-dependent control of arterial blood pressure.

## 4. Postcapillary Venules Hyperpermeability

The endothelial barrier controls nutrients and solute exchange between blood and tissues. However, during inflammation, ECs from postcapillary venules are susceptible to the disruption of their barrier and increased permeability to macromolecules, also called hyperpermeability [126,127]. The rearrangement in tight junctions (TJs), adherent junctions (AJs), and cytoskeletal alterations induce contractile forces leading to form gaps between ECs, so that macromolecules and even immune cells can pass through [128,129,130]. Takeuchi et al. showed that cell extrusion is driven by actomyosin contraction and triggered by a Ca^2+^ wave involving IP_3_ receptors, GJ, and TRPC1 [131]. In oncogenically transformed or apoptotic cells, this Ca^2+^ wave induces the polarized movement of the surrounding cells toward the extruding cells and facilitates apical extrusion by inducing actin rearrangement in the nearest neighboring cells (reviewed in [132]). Normally, the inflammatory environment is resolved and hyperpermeability is transient, contributing to repair and wound healing. However, prolonged hyperpermeability leads to the impairment of tissues and constitutes a hallmark of several inflammatory diseases such lung injury, ischemic stroke, and sepsis [133,134,135,136].

### 4.1. Connexin Proteins in Postcapillary Venules Hyperpermeability

Connexin protein expression is mainly described in large veins (Table 2). Cx40 and Cx43 are found in ECs from the vena cava, femoral veins, portal veins, and postcapillary venules; however, Cx37 is less detected [137,138,139,140,141]. These connexins are also found in ECs in in vitro models such as human saphenous vein culture, human umbilical vein endothelial cells (HUVEC), and primary cultures or cell lines from microvascular vessels [142,143,144]. In contrast, Cx37, Cx43, and Cx47 are commonly expressed in ECs vein valves [139,140,145]. Cx37 and Cx40 knockout mice show localized hemorrhages, but permeability is not affected in basal conditions. However, this model was not evaluated in the inflammatory response [80]. The Cx37-deficient mouse lacks vein valves [139] and the ECs-specific deletion of Cx40 induces an increase in leucocyte adhesion [146].

GJ channels could contribute to the hyperpermeability mechanism because of their proximity and interactions with other junction proteins [170,171,172,173]. GJ channels and a hemichannel opening could be involved in hyperpermeability because the development of hyperpermeability is associated with an agonist-induced increment in the endothelial intracellular calcium concentration ([Ca^2+^]_i_), and maintaining hyperpermeability over time depends on an extracellular Ca^2+^ influx [174,175,176,177,178,179,180], while ATP signaling is related to hyperpermeability mainly through purinergic receptor activation [181,182].

In addition, the relationship between Cxs and hyperpermeability depends on the vascular territory, including in vivo or in vitro models, and agonists used. For example, in the pulmonary barrier, studies show that during sepsis, lipopolysaccharide (LPS)-induced hyperpermeability is related to increased Cx43 expression, which is associated to TJs and AJs decreased protein expression, such as ZO-1, claudin, and vascular endothelial (VE)-cadherin, respectively, or increased Cx43 GJ channels improving the spread of signaling molecules or ions such as IP_3_ and Ca^2+^, which affects vascular hyperpermeability [152,153,154,161,183,184,185]. There is no consensus explaining the involved mechanisms, but Zhang and others, describe a Rho-associated protein kinase (Rock)1-myosin light chain (MLC)2 phosphorylation pathway that relates to contractile forces’ disruption or activation of transcription factor (Tcf)-4/β-catenin when glycoprotein osteopontin (OPN) is involved [153,154]. Interestingly, in Idiopathic Pulmonary Hypertension, asymmetric dimethylarginine (ADMA), a nitric oxide synthase inhibitor, increased, and there was a reduced expression and phosphorylation of Cx43 mediated by the NO–cyclic guanosine monophosphate (cGMP) pathway [186]. The contribution of Cx40 GJ channels in the hyperpermeability response is ambiguous. In the gun-shot-induced acute lung injury (ALI) model, TNF-α and IL-8 increased hyperpermeability by a Cx40 expression decrease [187,188]. However, in the acid (HCl)-induced ALI model, thrombin and PAF produced Cx40 GJ channels’ inhibition and genetic deficiency mitigated vascular endothelial permeability via the upregulation of Rock1, causing the subsequent phosphorylation of Myosin Phosphatase target subunit (MYPT)1 and MLC20 [189]. Furthermore, the specific Cx43 hemichannel blocking by the P5 mimetic peptide in acute respiratory distress syndrome (ARDS), associated with ALI, leads to a reduction of the alveolar infiltration of innate immune cells and protection against LPS-induced lung injury [190].

In the blood–brain barrier (BBB), Cx43 GJ channels’ contribution in familial cerebral cavernous malformations type III hyperpermeability is related to increasing its expression and debilitating TJs’ structure, because of its interaction with ZO-1 [191]. Moreover, BBB disruption caused by cerebral ischemia increased Cx43 phosphorylation [151]. Phagocytosis of myelin debris by microvascular endothelial cells (MECs) impaired TJs and Cx43 GJ channels after spinal cord injury (SCI) and led to increased permeability. Cx43 downregulation was found to reduce both the extravasation of intravenously injected FITC-BSA (bovine serum albumin) and the recruitment of neutrophil leukocytes into the injured cord [192,193]. Interestingly, Cx43 downregulation is also induced by SARS-CoV-2 Spike and is accompanied by decreased AJs proteins, resulting in hyperpermeability in primary brain vascular ECs, and is more pronounced in diabetic brain vascular ECs [194]. Since the endothelial barrier also depends on other cells, the perivascular organization of Cx43 GJ channels in astrocytes contributes to maintaining vascular communication and the BBB equilibrium [195]. Cx43 hemichannel participation was evaluated by the short-time exposure of Cx43 peptide Gap27, inhibiting BK-triggered [Ca^2+^]_i_ oscillations and BBB permeability in mouse brain ECs [147] and in human brain endothelial cells [159]. Multi-walled carbon nanotube induces BBB barrier disruption mainly by ATP release [31]. Oxygen/glucose deprivation decreases Ca^2+^ in rat brain capillary endothelial cells, inducing a Cx43 hemichannel opening and, subsequently, ATP release, contributing to BBB damage [164]. In the retinal barrier, ischemia-reperfusion (I/R) induces the upregulation of Cx43 and increases vascular leakage. However, high glucose-induced hyperpermeability in retinal ECs provokes Cx43 downregulation accompanied with ZO-1 and occludin downregulation, which is associated with Cx43 GJ channels uncoupling and a loss of interaction of the C-terminus of Cx43 with the second PDZ domain of ZO-1 [155,156,196].

In other inflammatory models, there is evidence that α1AMP-activated protein kinase (AMPK) protects against LPS-induced ECs barrier disruption. AMPK deficiency leads to a substantial loss of Cx43 in ECs and is correlated with TJs and AJs disruption [162]. Treatment with a Cx43 carboxyl-terminal mimetic peptide protects against thrombin-induced hyperpermeability by stabilizing ZO-1 and VE-cadherin [197]. Cxs hemichannel contribution to the ATP-related mechanism is described in acute inflammation, where Cx43 hemichannel-mediated ATP release from ECs mediates leucocyte recruitment during a foreign body response to sterile implants [198]. In addition, an in vitro study in human EaHy 926 cells report that in high-glucose and IL-1β/TNF-α conditions, Cx43 hemichannel activity increases and is associated with ATP release and the subsequent activation of purinergic receptors and a [Ca^2+^]_i_ increase [163]. In systemic inflammatory response syndrome (SIRS), TNF-α activates and promotes the Cx43 hemichannel opening in a Ca^2+^-dependent manner, leading to renal vascular permeability and mortality; however, in the presence of Gap19, a specific Cx43 hemichannel inhibitor, there is a protective effect. Interestingly, electrophysiological evidence demonstrated a fast stimulatory effect of TNF on the Cx43 hemichannel opening, which was linked to [Ca^2+^]_i_-dynamics [28].

In the context of atherosclerosis, Cx43 participation is related to leukocyte migration, which is part of the inflammatory condition where hyperpermeability is involved. Then, oxidized low-density lipoprotein (ox-LDL) induced an increase in Cx43, JAM-A, and VE-cadherin protein expression [158]. In addition, statin drug Simvastatin suppresses the TNF-α-induced inhibition of GJ channels’ activity in an in vitro ECs model, using HUVEC, by upregulating Cx37 and Cx40 expression but downregulating Cx43 [199]. Moreover, in line with leucocyte-ECs interactions, in a hamster cheek pouch, the TNF-α inflammatory effect on leucocyte adhesion is suppressed in ECs Cx43-deficient mice as well as by a pharmacological blockade of GJ [200]; thus, Cx43 is predominantly involved in atherosclerosis-related leucocyte adhesion, this is consistent with its contribution to hyperpermeability.

There is evidence of connexin contribution to the hyperpermeability response, inasmuch as agonist-induced eNOS translocation from the plasma membrane to cytosol and subsequent cytosolic NO production is crucial to agonist-induced hyperpermeability [126,201,202,203,204,205], influencing S-nitrosylation and the disassembly of AJs proteins and VASP [206,207,208,209,210,211,212,213]. The participation of GJ channels or Cxs-hemichannels in hyperpermeability may be related to a NO-cGMP-PKG pathway. This could possibly be linked to S-nitrosylation of connexin proteins. For instance, there is evidence that NO induces the upregulation of Cx40 NO-cGMP-PKG pathway-dependent alleviating cerebral vasospasm [214]. On the other hand, β-adrenergic cardiac stress and NO donors evoke S-nitrosylation of Cx43 proteins in mouse cardiac cells [15,27] and in Cx43 expressing Xenopus oocytes at cysteine 271 [15], respectively. Additionally, Cx43 is closely associated with eNOS in heart subsarcolemmal mitochondria [215], regulating Cx43 open probability by post-translational modifications. However, the mechanisms by which GJ channels or Cxs-hemichannels and Cx-S-nitrosylation contribute to hyperpermeability are not fully evaluated yet.

At the present time, few of these studies have raised the question of whether GJ channels may contribute to maintaining the normal ECs barrier. In fact, in normal conditions, monolayers of brain or pulmonary ECs cultures treated with Cx43 or Cx40 blockers lead to an unstable barrier [149]. This issue is discussed by Ange et al., 2020 [162]; however, there is no evidence regarding inflammatory conditions.

### 4.2. Pannexin Proteins in Postcapillary Venules Hyperpermeability

Pannexin channels are less documented in veins and venules (Table 2). Panx-1 is the most evaluated in inflammatory conditions [65,159,165,168,169]. The relationship between Panx-1 channels and the hyperpermeability response was first approached in HUVEC where the knockdown of Panx-1 blocks’ ATP release was induced by thrombin [168]. The Panx-1 channel openings may contribute to the BBB disruption by ATP release under mimicked ischemic stroke conditions by using Ca^2+^-free media in a BBB cell line or oxygen/glucose deprivation in a rat brain capillary endothelial cell line [159,164]. The increasing infiltration of leukocytes in mouse cerebral ischemic sections is blunted in Panx-1 inducible knockout mice [166]. Moreover, multi-walled carbon nanotube-induced hyperpermeability is a result of ATP release associated to the Panx-1 channel opening [31]. In lung ischemia reperfusion injury, endothelial-specific Panx-1 inducible knockout mice demonstrate a protective phenotype after I/R with reduced endothelial permeability, edema, and inflammation [167]. Studies in ECs postcapillary venules describe that Panx-1 channels are involved in TNF-α-induced hyperpermeability and leucocyte adhesion and migration in vivo, mediated by subsequent ATP hydrolyzation and purine signaling that could lead to a [Ca^2+^]_i_ increase by a transient receptor potential vanilloid 4 (TRPV4) channel opening [65,165,216]. In HUVEC, long exposure to TNF-α may lead to Panx-1 channel opening and a [Ca^2+^]_i_ increase and is associated with IL-1β production [169]. The Panx-1 channel contribution to the hyperpermeability response may be related to oxidative stress conditions or purinergic signaling [65,165,167]; however, NO signaling participation is not considered, since Panx-1 channel activation by S-nitrosylation is ambiguous [124,217,218] or NO seems to inhibit the Panx-1 channel opening in a cGMP-PKC pathway [219].

## 5. Neurovascular Coupling

Neurovascular coupling (NVC) is a mechanism that applies to changes in cerebral blood flow in response to the increase in neuronal activity to ensure the glucose and oxygen supply necessary for cerebral functions [220,221,222]. NVC depends on the coordinated cell communication between neurons, astrocytes, and microvascular cells (endothelium, SMCs, and/or pericytes). All of them, including the extracellular matrix (ECM) components, form a functional unit called the Neurovascular Unit (NVU) [223,224,225]. NVU plays a vital role in regulating vasomotor tone and vascular homeostasis in the brain [220,226,227]. Astrocyte cells are critical in NVC, functioning as transductors between neurons and brain blood vessels through specialized structures called astrocytic endfeet, allowing several vasoactive mediators (including ATP, glutamate, D-serine, or other neurotransmitters) to enter the perivascular space [32,220,228,229,230,231]. This mediator release depends on the intracellular Ca^2+^ increase propagated as Ca^2+^ waves between neighboring astrocytes, and is coordinated by adenosine triphosphate (ATP) release [220,232,233,234,235,236,237,238,239].

### 5.1. NVC Mediators in Astrocytes and Vascular Cells

Neuronal activity is especially associated with glutamate, the main excitatory neurotransmitter in the central nervous system (CNS) [240,241,242]. Glutamate can activate ionotropic and metabotropic receptors in astrocytes [243,244,245,246], even though astrocytes express receptors to many neurotransmitters. Group I metabotropic glutamate receptors’ (mGluR) activation, especially mGluR5 expressed in astrocytes, correlates with intracellular Ca^2+^ oscillations observed during NVC through the phospholipase C (PLC) and inositol triphosphate (IP_3_) signaling [240,247,248]. Interestingly, mGluR5 expression in astrocytes was reported to decrease during development, being absent in the adult; thus the role of astrocytic mGluR5 in NVC might depend on developmental age (as reviewed in [249]). Arachidonic acid (AA) production under the phospholipase C A2 (PLCA2) signaling pathway in astrocytes constitutes the substrate of cytochrome P450 epoxygenases (CYP450) or cyclooxygenase (COX), allowing the synthesis of SMCs vasodilators such as EET and prostaglandins (PGs), respectively [230,250,251].

Astrocyte studies using primary cultures, brain slices, and in vivo murine models have shown an autocrine role of EET, triggering Ca^2+^-activated potassium channels (K_Ca_), increasing the intracellular Ca^2+^, and as a paracrine signal in SMCs, producing hyperpolarization and vasodilation [251,252,253,254]. This response is activated after EET binding to G-protein coupled receptors (GPCR) expressed in SMC brain vasculature, and the large conductance of Ca^2+^-activated potassium channel (BK) activation [255,256]. Furthermore, it has been reported that prostaglandin E2 (PGE-2) correlates with parenchymal vasodilation after direct astrocytic stimulation in a rat cortical slices model [251,257,258]. The activation of the E-prostanoid receptors family, probably EP_4_, coupled to adenylyl cyclase via G_s_-proteins has been related to SMCs’ relaxation in the brain during NVC [259,260,261].

K^+^ released from the astrocyte endfeet is an essential astrocyte signaling ion associated with vasodilation in brain blood vessels. The astrocytic K_Ca_ allows a K^+^ increase in the perivascular space, activating inwardly rectifying K^+^ (K_ir_) channels in brain capillaries/arterioles, inducing hyperpolarization and conducting vasodilation so that feed arteries provide sufficient blood to satisfy the metabolic demand [220]. Nevertheless, the K^+^ perivascular concentration can determine switches between vasodilation and vasoconstriction [235,260,262,263]. The production of 20-hydroxyeicosatetraenoic acid downstream of the AA pathway in astrocytes has been related to the vasoconstriction of parenchymal brain vessels [258,263,264,265,266]. These observations show the complex and regulated interactive signaling produced during the astrocytic Ca^2+^ increase associated with the AA metabolism and K^+^ released from astrocytes during NVC. 

Another essential component during NVC is the astrocytic ATP release, which coordinates Ca^2+^ waves in an autocrine manner and has vasomotor activity effects [233,261,267,268,269]. ATP activates metabotropic purinergic receptors (P2Y) evoking an IP_3_ signaling pathway in astrocytes [268,269], and it has also been observed in pericytes and brain SMCs, capillaries, and pial arterioles, where the P2Y receptors activation (and apparently not through P2X) leads to an intracellular Ca^2+^ increase and vasoconstriction [268,270]. Additionally, in vitro and in vivo experiments using SMCs from rat brain arterioles suggest coupling between TRPV4 and P2Y receptors, leading to the constriction of cerebral parenchymal arterioles [271]. However, due to the short half-life of ATP (by the extensive expression of its hydrolyzing ectonucleotidases), the potent brain vasculature vasodilator adenosine production is consistent with the responses mediated by astrocytes during NVC [239,272,273,274,275]. Brain ectonucleotidases are expressed in microglia, ECs, SMCs, and astrocytes membranes, permitting to consolidate an adenosine source in the brain [276,277,278].

There are four types of adenosine receptors: A1, A2A, A2B, and A3. A1/A3 activate inhibitory signaling through Gi-PLC signaling, while A2A/A2B is associated with an activating pathway through Gs-cAMP [279]. Studies in isolated brain arterioles have shown the association between vasodilation and A2A/A2B receptors’ activation [280,281]. SMCs in cerebral arteries and arterioles express adenosine receptors A2B, activating the GPCR adenylate cyclase/cAMP/PKA pathway and K_ATP_ channels, which hyperpolarize the SMCs [282,283]. This pathway has been associated with vasodilation in brain pial arterioles during in vivo and in vitro studies, in ECs’ nitric oxide production-dependent manner, under adenosine receptors’ activation [284,285,286].

Despite the relevance of ATP release from astrocytes during NVC, there is no consensus about the ATP release mechanism. Several channels have been proposed in purine release and their possible contribution to the intracellular Ca^2+^ signaling pathway in NVC. Thus, the Cxs hemichannels and the Panxs channels are promising candidates due to their permeation properties and expression patterns in astrocytes [287].

### 5.2. Cxs Expression in Astrocytes and Vascular Brain Cells, and Possible Role in NVC

Astrocytes express Cx43 predominantly; fewer levels of Cx26, Cx30, Cx40, and Cx45 have been described as hemichannels, while astrocyte GJ channels are formed by Cx26, Cx30, and Cx43 [288,289,290,291,292,293]. The high coupling between astrocytes through Cx43 GJ channels permits Ca^2+^ propagation in the astrocytic network, developing an extensive brain territory signaling coordinated by ATP release [294,295,296]. However, whether Cx26, Cx30, and Cx43 hemichannels contribute to an intracellular Ca^2+^ influx directly or by ATP release (which activates purinergic receptors) during NVC has not been completely established. Based on two observations, it is possible to hypothesize that Cxs hemichannels participate in the astrocytic Ca^2+^ signaling during NVC: (1) the inhibition of the intracellular Ca^2+^ signaling using GPCR blockers does not abolish the Ca^2+^ increase in astrocytes altogether [297], which suggests that the dominant Ca^2+^ entry is via another Ca^2+^-permeable channel, and (2) an intracellular Ca^2+^ increase in astrocytes mediate Cx43 hemichannels opening [298,299]. Furthermore, a Cx43 hemichannel-mediated ATP release has been described in astrocytic cell lines and primary cultures, associating it to the intracellular Ca^2+^ waves by P2Y receptors’ activation [298,300,301]. However, these events have not been completely elucidated in NVC.

The Cx37/Cx40/Cx43 expression in ECs and SMCs has been reported in brain basilar arteries and pial branches [302,303]. Cx45 is only expressed in brain SMCs arterioles [303,304], and Cx37/Cx40 have been identified in myoendothelial junctions [302,305]. The Cxs expression in ECs and SMCs in brain blood vessels acquires relevance according to the study by Figueroa et al., where NO diffusion through Cx43, Cx40, and Cx37 hemichannels and myoendothelial GJ channels was established [85]. NO diffusion through these channels could correlate with NO signaling in the NVC. Nevertheless, it is unknown if these events are involved in vasodilation during NVC.

In addition to the Cxs expression in astrocytes and vascular brain cells, Panx-1 and Panx-2 are abundantly expressed in many regions of the CNS [306,307], without a defined role in NVC. 

### 5.3. Panx-1 Expression in Astrocytes and Vascular Brain Cells, and Possible Role in NVC

Only Panx-1 expression has been consistently reported in astrocytes [308,309,310]. In vitro studies, using mice astrocytic primary cultures, have shown that Panx-1 channels can release ATP after depolarization or the downstream of P2X7R activation [311,312]. The participation of P2XRs in NVC has not been entirely accepted. It is possible that the high extracellular concentration of the P2X7R agonist, BzATP, used in these studies may permit these receptors’ activation, which do not participate under physiological conditions. Nonetheless, these studies support the possible release of ATP from astrocytes through Panx-1 channels [313], and establish it as a candidate for this signaling during NVC.

Consistent with Girouard et al., 2010, studies, where the extracellular K^+^ concentration in the perivascular space regulates the vasomotor responses in murine brains, Scemes and Spray showed that the increase of extracellular K^+^ observed during NVC is associated with an increase in the astrocytic coupling through Cxs GJ channels, leading to the activation of Panx-1 channels and ATP release from astrocytes [314], which is in line with the observations during NVC, and could establish the differential contribution of these types of channels in this mechanism.

Finally, Panx-1 is expressed in ECs and SMCs from mice cerebral arteries. According to recent observations, ECs regulate the myogenic tone in small brain vessels in a P2YRs-dependent manner [166]. In the same study, Good et al. found that endothelial Panx-1 did not control the myogenic tone in small peripheral vessels. These data suggest the territorial-space activity of Panx-1 channels mediating the arterial vasomotor tone. These functional activities of Panx-1 channels may be considered to determine whether pannexin proteins are cardiovascular targets to prevent/improve vascular dysfunction.

## 6. Conclusions

Cxs and Panxs channel activity and their differential expression in distinct cells of the blood vessels have a crucial role in the physiological and pathophysiological processes associated with vascular homeostasis. Cxs and Panxs contribute to microcirculatory regulation as their participation has been observed during the vasomotor control exerted by arterioles as well as during hyperpermeability (at postcapillary venules) observed under inflammatory conditions. We discussed also neurovascular coupling, an essential mechanism for adequate brain function, where primarily Cx43 and Panx-1 participate actively in mediating complex signaling pathways associated with NVC regulation.

The role and participation of Cxs and Panxs large-pore channels are still under experimental observation. For instance, recent results display a transport behavior with a molecular permeability at a negative potential (i.e., resting membrane potential) in the absence or reduced atomic ion conductance [6]. The physiological and pathophysiological roles of Cxs and Panxs could be essential in homeostatic functions of the microcirculation.

We believe Cxs and Panxs might be considered a robust therapeutic target for controlling physiological and pathophysiological functions in microcirculation and NVC. However, mechanistic studies and novel technical approaches are required to establish firmly the physiological role of these large-pore channels, which presents a potentially fruitful line of research to be addressed in the near future.

## Figures and Tables

**Figure 1 ijms-23-07303-f001:**
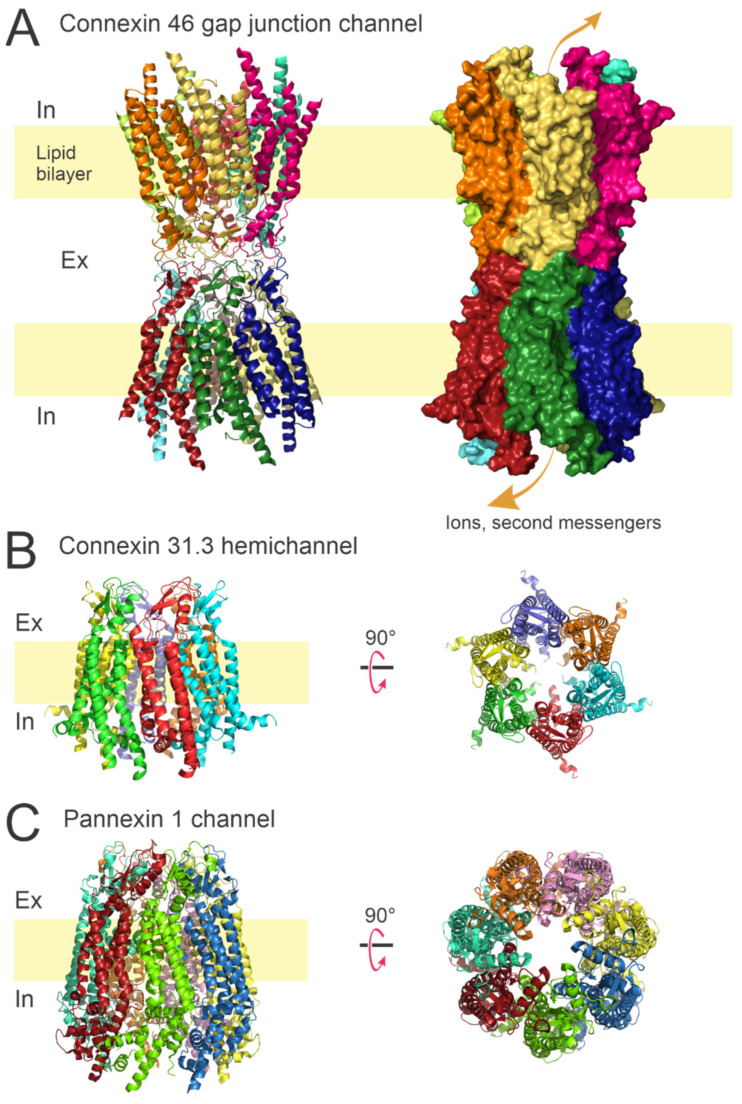
Overall architecture of gap junction channels, connexin hemichannels, and Panx-1 channels. (**A**) Ribbon and surface representations of the Cx46 GJ channel viewed from the lateral side. The dodecameric assembly allows the permeation of ions and small metabolites between adjacent cells. (**B**) Ribbon model for the Cx31.3 hemichannel viewed from lateral side (**left**) and from the extracellular side (**right**). Hemichannels assemble in hexamers and provide a conduction pathway for ions and small molecules between extracellular and intracellular compartments. (**C**) Ribbon representations of the Panx-1 channel. Panx-1 channels assemble as heptamers and provide a pathway for paracrine/autocrine signaling. Protein data bank (PDB) codes: 7JKC (Cx46) [36], 6L3U (Cx31.3) [37], and 6WBF (Panx-1) [38].

**Table 1 ijms-23-07303-t001:** Connexin and Pannexin protein distribution in arterioles (+ expression/− no expression).

Protein	Species	Blood Vessel	Endothelial Cell	Smooth Muscle Cell	Reference
	Rat	Coronary artery	+	+	[66]
Cx37		Arterial mesenteric bed	+	−	[67]
	Mouse	Cremaster muscle	+	−	[12]
	Hamster	Cheek pouch arterioles, retractor muscle feed arteries	+	+	[67]
	Pig	Coronary arteries	+	−	[66]
	Rat	Coronary arteries	+	+	[66]
Cx40		Arterial mesenteric bed	+	−	[67]
		Brain Pial arterioles	+	+	[67]
	Mouse	Cremaster muscle	+	+	[12,68]
	Hamster	Cheek pouch arterioles, retractor muscle feed arteries	+	−	[67]
	Pig	Coronary artery	+	+	[66]
	Cow	Coronary artery	+	+	[66]
Cx43	Rat	Coronary artery	−	+	[66]
		Arterial mesenteric bed	+	−	[67]
	Mouse	Cremaster muscle	+	+	[12,68]
	Hamster	Cheek pouch arterioles, retractor muscle feed arteries	+	−	[67]
Panx-1	Rat	Arterial mesenteric bed	+	+	[69,70]
	Mouse	Pinotrapezius muscle and cremaster muscle arterioles	+	+	[71]
		Coronary arteries	+	−	[71]
		Abdominal arteries	+	+	[71]
Panx-3	Mouse	Coronary arteries	+	+	[71]
		Cortical kidney arterioles	+	−	[71]

**Table 2 ijms-23-07303-t002:** Connexin and Pannexin protein distribution in in vivo veins, venules, and in vitro cell models (+ expression/− not expression).

Protein	Species	Blood Vessel or Cell Culture	Endothelial Cell	Valves	Inflammatory Model	Reference
Cx37	Rat	Vena cava	+	−	−	[137,138]
		Rat brain endothelial (RBE4)	+	−	Bradykinin acute inflammation	[147]
		Pulmonary microvascular endothelial cells (PMVEC)	+	−	In vitro acute lung injury	[141]
	Mouse	Vena cava, Femoral vein, Saphenous vein, Braquial vein	−	+	−	[139,140,145]
		Portal vein	+	−	Mouse liver development	[148]
	Human	Saphenous vein	+	−	In vitro vein hyperplasia, cryopreservation	[142,143,144]
		Saphenous vein	−	+	−	[145]
	Rabbit	Yugular vein	+	−	−	[144]
	Pig	Primary culture of porcine brain microvascular endothelial cells	+	-	−	[149]
Cx40	Rat	Vena cava	+	−	-	[137,138]
		Rat brain endothelial (RBE4)	+	−	Bradykinin acute inflammation	[147]
		Pulmonary microvascular endothelial cells (PMVEC)	+	−	In vitro acute lung injury	[141]
	Mouse	Vena cava, Femoral vein, Saphenous vein, Braquial vein	−	+	−	[139,140,145]
		Portal vein	+	−	Mouse liver development	[148]
		Renal Interlobular veins	+	−	−	[150]
	Human	Saphenous vein	+	−	In vitro vein hyperplasia	[142,143,144]
		Saphenous vein	−	+	−	[145]
	Rabbit	Yugular vein	+	−	−	[144]
	Pig	Primary culture of porcine brain microvascular endothelial cells	+	−	−	[149]
Cx43	Rat	Vena cava	+	−	−	[137,138]
		Rat brain endothelial (RBE4)	+	−	Bradykinin acute inflammation	[147]
		Rat Brain microvascular endothelial cells (RBMEC)	+	−	Thrombolysis	[151]
		Pulmonary venules	+	−	Endotoxin-challenged lungs	[152]
		Pulmonary vein endothelial cells	+	−	Sepsis	[153,154]
		Retinal endothelial cells	+	−	In vitro diabetic retinopathy	[155,156]
		Pulmonary microvascular endothelial cells (PMVEC)	+	−	In vitro acute lung injury	[141]
	Mouse	Vena cava, Femoral vein, Saphenous vein, Braquial vein	−	+	−	[139,140,145]
		Portal vein	+	−	Mouse liver development	[148]
		Retinal vein	+	−	−	[157]
	Human	Saphenous vein	+	−	In vitro vein hyperplasia	[142,143,144]
		Saphenous vein	−	+	−	[145]
		HUVEC	+	−	In vitro atherosclerosis	[158]
		Human microvascular endothelial cell line (hCMVEC)	+	−	In vitro Ischemia injury and Extracellular Ca^2+^ variations	[159,160]
		Human pulmonary microvascular endothelial cells	+	−	Acute lung injury	[161]
		Human dermal microvascular endothelial cells (HMEC)	+	−	In vitro Sepsis	[162]
		EAhy Cells	+	−	Inflammatory cytokines (TNF-α, IL-8) and high glucose exposure	[28,163]
	Rabbit	Yugular vein	+	−	−	[144]
	Pig	Primary culture of porcine brain microvascular endothelial cells	+	−	−	[149]
Cx47	Mouse	Vena cava, Femoral vein, Saphenous vein, Braquial vein	−	+	Vein development	[139,140,145]
	Human	Saphenous vein	+	+	In vitro vein hyperplasia, vein development	[142,145]
Protein	Species	Blood Vessel or Cell Culture	Endothelial Cell	Valves	Inflammatory Condition	Reference
Panx-1	Rat	Rat Brain Capillary endothelial cells TR-BBB13	+	−	In vitro brain ischemia	[164]
	Mouse	Mesenteric postcapillary venules	+	−	TNF-α acute inflammation	[65,165,166,167]
	Human	HUVEC	+	−	Thrombin, TNF-α inflammation	[165,168,169]
		Human microvascular endothelial cell line (hCMVEC)	+	−	Extracellular Ca^2+^ variations	[159,164]
		Primary Human Saphenous vein Endothelial cells	+	−	TNF-α inflammation	[165]

## Data Availability

Not applicable.

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
