# Peer review of "Connexin and Pannexin Large-Pore Channels in Microcirculation and Neurovascular Coupling Function"

_ijms, 2022, doi:10.3390/ijms23137303_

Round 1
Reviewer 1 Report
Please find my comments attached

Author Response
Dear Review, thanks for your comments and feedback. We included all your comments and corrections in our manuscript. These changes are in track changes. Besides, we added a new paragraph in yellow in the word attached.
Please see the attachment.
Mauricio Lillo, PhD

Reviewer 2 Report
Excellent, comprehensive review article which discusses the role of connexin and pannexin channels in the regulation of the microcirculation and in neurovascular coupling. One issue that I have is that the section on Connexin proteins in arterioles is confusing. It would be helpful to discuss the mechanisms by which gap junctions and connexin hemichannels are thought to regulate vascular tone in more detail.
Author Response
Dear Review, thanks for your comments and feedback. We included new information highlighted in yellow, describing better the potential role of connexin proteins regulating vasomotor tone. In addition, we added all the suggestions by Reviewers 1 and 3.
These changes are in track changes. Besides, we added a new paragraph in yellow in the word attached.
Best,
Mauricio Lillo, PhD

Reviewer 3 Report
The review is very detailed and well written. it completely covers the subject. There are only a few small things to consider: Microcirculation could be added as a keyword Page 1: "small molecules" passing through gap junction channels could be more defined (size and hydrophilic) Page 3: In the pannexin channels paragraph, rewrite: "in adjacent cells probably because of several glycosylations..." and in the second paragraph of this part: "in several cell types and tissues". In the last sentence of this paragraph, add a dot for "i.e." Page 5: Consider the order of the different parts. Is the part "Pannexin proteins in arterioles" really the part "2.1" when "Connexin proteins in arterioles" is "2"? And afterwards, consider the following subtitles if they are correctly written. In general, it seems there is a lack of harmonization in writing such as: -Ca2+ that should be written correctly (2+ as exponent) in different parts of the text (Abstract; page 2; Table 2, pages 8 and 9) -In vitro/in vivo in italic or not in the text but choose one way. -The Cx, Panx and GJ abbreviations could be more generally used in the text of the manuscript -There are some long spaces (pages 2, 5, 6 -In Table 2, please harmonize "vein" or "Vein". Both types are in the table. "Jugular vein but not yugular vein. In vitro/in vivo is in italic or not. Page 8, in the Table 1: No species for Eahy cells (human?) Page 8, in the Table 2, please write "and extracellular" Page 10 (2nd paragraph): write "carboxyl", remove dot after [180], and coma after "implants". Page 11, in part 2.4, remove coma after "cell line". Page 12: Write IP3 (3 as a subscript) in 1st and 4th paragraphs of part 3.1; and in the 2nd paragraph, write "activated" potassium channels. 3rd paragraph, remove the dot after "demand". Page 14 (1st line): "establish" Page 14: Harmonize police writting of the ConclusionAuthor Response
Dear Review, thanks for your comments and feedback. We incorporated all your comments and other editions and suggestions by reviewers 1 and 2.
These changes are in track changes. Besides, we added a new paragraph in yellow highlighted in our new word attached.
Please see the attachment.
Best,
Mauricio Lillo, PhD
